# Neural Architecture Search for 1D CNNs—Different Approaches Tests and Measurements

**DOI:** 10.3390/s21237990

**Published:** 2021-11-30

**Authors:** João Rala Cordeiro, António Raimundo, Octavian Postolache, Pedro Sebastião

**Affiliations:** 1Instituto de Telecomunicações (IT-IUL), Instituto Universitário de Lisboa (ISCTE-IUL), 1649-026 Lisbon, Portugal; lima_raimundo@iscte-iul.pt (A.R.); Octavian.Adrian.Postolache@iscte-iul.pt (O.P.); 2Department of Information Science and Technology, Instituto Universitário de Lisboa (ISCTE-IUL), 1649-026 Lisbon, Portugal; pedro.sebastiao@iscte-iul.pt

**Keywords:** Neural Architecture Search, 1D CNN, 1D NAS, CNN hyperparameters, CNN architecture tuning, tests and measurements, optimisation algorithms testing

## Abstract

In the field of sensors, in areas such as industrial, clinical, or environment, it is common to find one dimensional (1D) formatted data (e.g., electrocardiogram, temperature, power consumption). A very promising technique for modelling this information is the use of One Dimensional Convolutional Neural Networks (1D CNN), which introduces a new challenge, namely how to define the best architecture for a 1D CNN. This manuscript addresses the concept of One Dimensional Neural Architecture Search (1D NAS), an approach that automates the search for the best combination of Neuronal Networks hyperparameters (model architecture), including both structural and training hyperparameters, for optimising 1D CNNs. This work includes the implementation of search processes for 1D CNN architectures based on five strategies: greedy, random, Bayesian, hyperband, and genetic approaches to perform, collect, and analyse the results obtained by each strategy scenario. For the analysis, we conducted 125 experiments, followed by a thorough evaluation from multiple perspectives, including the best-performing model in terms of accuracy, consistency, variability, total running time, and computational resource consumption. Finally, by presenting the optimised 1D CNN architecture, the results for the manuscript’s research question (a real-life clinical case) were provided.

## 1. Introduction

In the past years, the field of sensors, smart sensors, artificial intelligence, and the combination of these fields has made enormous progress. Specifically, in the machine learning area, Artificial Neural Networks (ANN), Deep Neural Networks (DNN) (the most common techniques in deep learning field), and Convolutional Neural Networks (CNN) are concepts in vogue, reflecting a growing scientific interest and the consequent contribution from the scientific community [1,2,3].

CNN is a specific class of deep neural networks and perhaps the most popular algorithm among the deep learning environments. To understand this popularity, there is a need to understand the deep learning and CNN’s main concepts. Since the emergence of deep learning-based algorithms, the traditional machine learning techniques were limited to process data in raw format, which required careful feature engineering and extraction expertise to transform natural data into tabular forms (handcrafted features space). This process can achieve levels of complexity that often lead to information loss. Deep learning applied representation learning methods that allow models to discover representations from natural data, such as pixel values in an image, replacing the handcraft and “imperfect” human data transformation [4,5].

The deep learning concept was first introduced in 2006 by a group of researchers of the Canadian Institute for Advanced Research (CIFAR). It refers to models composed by multiple processing layers, and each layer can learn representations with multiple levels of abstraction, capturing the full complexity within the data. The existence of several layers in the network leads to the origin of the “deep” term when referring to this type of models [5,6,7].

Deep learning has recently been used successfully in many fields such as visual object recognition, speech and audio processing, natural language processing, object detection, among others [7,8,9,10].

### 1.1. Convolutional Neural Networks

A Convolutional Neural Network, specifically, is a deep feed-forward network that extends a “classic” artificial neural network by adding more layers (deep learning), including the introduction of convolutions blocks. The term “convolutional” comes from the implementation of convolution blocks in the network. The first appearance of this type of neural networks dates to 1980, when Kunihiko Fukushima introduced the concept of convolution and down sampling layers, and to 1990, when LeCun et al. published a paper presenting the principles of the CNN modern framework [11,12,13].

The typical structure of a CNN is composed of three layers: convolutional, pooling, and fully connected layers. These layers make use of several components/techniques, such as convolutions, activation functions, pooling, dropout, batch normalisation, fully connected blocks, among others, that can be combined in multiple ways. A CNN structure can be built according to the following aspects [11,13,14]:The input data can have a 1D, 2D, or 3D format. The source of this data can be, for instance, from sensors, audio, video, and 3D images.The convolutional layers are responsible for processing the feature extraction tasks. It processes the extraction by applying convolution operations to the input, and the convolution result is fed to the next layer’s input. The convolutions operations are defined by several filters, a kernel size, a padding, and a stride, which finally generates a feature map after applying an activation function, such as ReLU or Tanh function.The pooling layers are usually applied after a convolutional layer, and they are responsible for holding the information generated by the feature maps. For instance, when processing an image, these layers drastically reduce the size of the input, speeding up computation times, making the training process faster and consequently, making a more robust feature detection. The most common techniques applied on these layers are max pooling and average pooling.The fully connected layer is a classic backpropagation neural network, which processes the features generated by previous layers. It produces the network final output, a prediction, which can be a regressing task, e.g., a metric forecast, or a classification task, such as classifying an image in classes.

Considering the previously described layers, it is important to give a brief explanation of the most relevant CNN components:Kernel: In each of the convolution layers, the input is run/slid by a kernel. A kernel consists of mathematical operations that produce a matrix dot product called an activation map. This operation is processed several times (defined by the number of filters) on the same input, using different kernels. Each filter with the associated kernel will extract different features from the input.For example, let us consider a simple case where the first layer of a 2D CNN has as input a grayscale image with a 28 × 28 resolution. The input will be a 2D matrix where each element represents a pixel value (784 pixels). The kernel size will define the kernel matrix dimension, which consists of a matrix of weights. The most popular kernel sizes are 3 × 3 or 5 × 5, which are initialized via initialization procedures such as Xavier or MSRA. Each pixel with be multiplied by the kernel matrix, generating an activation map and, usually, several filters will be used, resulting in several activation maps by layer. The way these maps are generated (and the corresponding size) also depends on stride and padding settings.Stride: The stride value defines how the kernel moves in the input data. The most common value is 1, meaning the kernel moves over input one column at each time. If the value is 2, the kernel moves two columns, and so on. The stride can take different values to extract different kinds of features.Padding: Padding is a technique that avoids the loss of information on the input data borders, caused by the kernel operations, and consists of adding zeros around the input margins. Considering the grey image example and a kernel size of 3 × 3: without padding, the first pixel would be processed only once, the second pixel would be processed twice, while some pixels (the ones out of the image border) would be processed nine times. The padding gives the opportunity for all pixels to be equally considered, improving the model learning process.Activation Function: The activation functions are responsible for applying non-linearity to the models, through the application of the operations on activation maps, making it possible to incorporate complex input relations. To achieve the goal, the activation maps can be submitted to the activation functions, generating feature maps. This operation can be performed by several activation functions such as Sigmoid or ReLU, which differ in their behaviour and consequently on the generated output. For instance, a Sigmoid will consider the activation map individual values and output 0 or 1, where a ReLU will result in the same value for positive values and 0 for negative values, which will generate different features.Pooling: Regarding the pooling operations, the goal is always to reduce the spatial size of the convolved features and avoid overfitting. There are several ways of processing pooling, which consists of different mathematical operations, processed similarly to the kernel operations. For instance, max pooling consists of running the input with a defined spatial neighbourhood and stride, taking the max value from the considered region. The average pooling consists of the same process, replacing the max by the average. In short, it delivers a summary statistic of the nearby outputs.Dropout: Dropout is a technique used for avoiding overfitting, which can be applied to fully connected or convolutional layers. It consists in randomly nullifying some of the connections on the neural network, which implies that individual neurons can be disposed and those are prevented to acquire excessive importance on the model output. In another way, the technique improves the model generalisation.

### 1.2. The Use of CNNs

The use of CNNs has dramatically improved the state of the art on image processing (including image classification, image semantic segmentation, and object detection) and nowadays, is still the main reason for their popularity, having many successful contributions among diverse areas such as agriculture, healthcare, transportation, and weather forecast. Good examples of CNN applications are presented by A. Kamilaris and F. Prenafeta Boldú on farming disease detection and fruit counting, or by Xiaolei Ma et al., who apply CNNs to traffic prediction in the transportation area [11,15,16,17].

Besides the application of CNNs on image scenarios (data in 2D format), with several popular architectures such as AlexNet, VGG, GoogleNet, ResNet, and Inception, and worldwide image processing competitions (e.g., ILSVRC—ImageNet Large Scale Visual Recognition Challenge and VOC—PASCAL Visual Object Classes), CNNs structures make it possible to process other types of data, such as natural language, video, speech, or audio [18,19,20,21,22].

Though the flexibility of the CNNs, given by 1D, 2D, or 3D convolutions, and the widespread use of 2D convolutions, it is not common to find 3D convolution models, which probably can be explained by less availability of 3D data. One dimensional convolution models, despite the availability of 1D data, also have a lower popularity [23,24,25].

There are even situations where 1D data has been converted/reshaped to 2D format, so 2D CNNs can be applied. On the recent work about air quality presented by Mirche Arsov et al., with the title “Multi-Horizon Air Pollution Forecasting with Deep Neural Networks”, the authors reshaped 1D data to 2D in order to apply a 2D CNN to their work [21,26].

Despite the scarcity of 1D CNN research, some very recent research works published on MDPI Sensors Journal can be found, e.g., the work developed by R. A. Osman et al., on interference avoidance on device communications using 5G communication; the advanced diagnosis of soft faults on motor power cables by H. Kim et al.; the research “Estimation of Tool Wear and Surface Roughness Development Using Deep Learning and Sensors Fusion” presented by P.M. Huang and C.H. Lee; the monitoring of the spatter behaviours on acoustic signals developed by S. Luo et al. Although all the papers cited before applied 1D CNNs on their research, neither of these very recent works used Neural Architecture Search (NAS) techniques (some do not even discuss the optimisation), resorting instead to the “traditional” grid search or trial-and-error approaches, which is the research gap that led to the need for this research [14,27,28,29,30].

There are several advantages in using 1D CNNs, such as good performances on a limited amount of data, low computational complexity (when compared with 2D CNNs and other deep learning architectures), faster training process, and a good ability to extract relevant features in sequence data and time-series (e.g., signal data). Serkan Kiranyaz et al. in the article “1D convolutional neural networks and applications: A survey” discuss in detail these advantages. The research works previously presented as state of the art are also good examples on how to benefit from the implementation of 1D CNNs [25,27].

Nevertheless, the number of published articles on 1D and 3D convolutions is pretty narrow when compared with 2D, and, to the best of our knowledge, there are no reference architectures for any of the cases. There are great opportunities to explore 1D and 3D CNN in areas such as: medical 3D images (like computed tomography, magnetic resonance imaging, or ultrasound) and object identification or segmentation; 1D signals, such as electrocardiograms (ECG), electroencephalograms (EEG) and electromyograms (EMG) on healthcare applications; and, e.g., on machine fault detection. In these areas, there is a lot of data available, which allows further research on different CNN applications [21,23].

### 1.3. The Need for Optimised CNNs

The success of a CNN, and in general for any neural network, depends on finding an architecture to fit a given problem. The most straightforward approaches are to use an existing network structure (e.g., a reference network), adapt that structure, or build and configure a new architecture. Any of these cases does not guarantee the best solution for a given problem (and it can be easily far from the optimal solution), but architecture optimisation is a challenging, complex, and time-consuming process. Moreover, when building and configuring a new architecture from scratch, it can be extremely difficult to achieve the optimal solution, due to the many architectural design options that require a specific level of expertise, knowledge, and effort [31].

A CNN architecture (depicted in Figure 1) is defined by a set of hyperparameters, namely structural and training hyperparameters. The structure hyperparameters are the number of layers, number of units in each layer, kernel size, number of filters, stride, pooling, normalisation techniques, and activation functions. The training hyperparameters are the optimiser, learning rate, momentum, batch size, initial weights, epochs, and patience (for early stopping method) [31,32].

The performance of a CNN is critically sensitive to the architectural design and differs for each problem/challenge, leading to a difficult task of finding the appropriate hyperparameter values and their combination. It becomes even harder to understand network hyperparameters relations and their impact on the model’s performance. Since there is no mathematical formula for calculating the appropriate hyperparameters combination, it is common to find the grid search or random search as strategies to optimise the architecture. These strategies present some drawbacks, such as high usage of computational resources and non-directional (“dummy”) search for the optimal solution. Even with these strategies, there is also no guarantee that the resulting model is the best solution. More recently, Bayesian optimisation methods and genetic algorithms have been considered by the scientific community as approaches to optimise the architecture, but still with quite limited experiments [31,33].

### 1.4. Neural Architecture Search for Optimising Neural Networks 

Figure 2 illustrates the NAS concept that describes the process of automating the search for the optimised design for neural networks through a search strategy. This concept was introduced in 2017 by Research at Google Zoph and Le 2017 [34,35].

As previously mentioned, a great amount of the existing work in deep learning is focused on 2D CNN and, thus, the NAS work is mainly focused on image processing with the use of 2D CNNs. There are also some research works that have started to emerge, e.g., on semantic segmentation, deep reinforcement learning, and machine translation [35,36,37].

The work described in this paper started from a real need from a research project. The project included the need to implement a performant machine learning model for the classification of a 1D signal. This project’s use case consists in exploring a clinical scenario where ECG waves needed to be collected and then, by implementing a classifier, classify those waves. The CNNs were a promising solution, but there was the need to find the best architecture to solve this classification problem. The literature review has revealed some strategies, such as the application of trial-and-error, greedy, random, and Bayesian approaches, but to the best of our knowledge, there were no libraries or frameworks able to process the 1D CNN architecture optimisation, neither comparison between approaches which could help the research team on the work that needed to be done.

To address this gap, the team has developed NAS processes for 1D CNN architectures and developed the comparison between the five different search strategies: greedy, random, Bayesian, hyperband, and genetic, in order to understand which strategy best fits the requirement of achieving an optimised architecture.

The NAS processes, developments, and implementation details on 1D CNN have been shared with the scientific community. When applying these scientific contributions on professional or academic use cases, 1D CNN researchers that develop their work based on exploring 1D CNN architectures can benefit from the conclusions achieved in this work.

The current paper offers a detailed view on how the five strategies behave, and it also shows the process of finding the best 1D CNN architecture based on performance stability, running time, and computational resources. This information is very useful, allowing researchers and developers to make better decisions when facing the need to use a NAS process for a 1D CNN problem.

In short, there is a huge potential in the use of 1D CNN but currently, researchers seem to avoid 1D CNN due to lack of expertise or use unoptimised 1D CNN due to lack of optimisation tools. This paper presents the implementation of NAS approaches for 1D CNNs, as well as a search strategies comparison analysis, which certainly will benefit the community on 1D data modelling with the use of optimised 1D CNNs.

## 2. Materials and Methods

In order to compare the different NAS approaches, it was decided to carry out a methodology to develop a structured and non-bias testing and measurement method. To achieve that, the benchmark approach was used.

Benchmark, in a general way, can be described as the comparison of one or more products over a series of measures, which can be applied to several scientific and business areas. In the comparison of the architecture optimisation algorithms, the products are the specific implementations of the given algorithms, and the performance metrics are obtained by running the implementations on the test sets [38].

It is quite common to find structured methods or processes on business companies, e.g., for comparing, assessing, and/or measuring different perspectives such as product offers or sales, personal performance, or financial ratios. Nevertheless, it becomes difficult to find processes or best practices for optimisation algorithm comparison. After a thorough literature review, the decision was based on the recent framework presented by Vahid Beiranvand et al. in the article “Best Practices for Comparing Optimisation Algorithms”, which was created for helping researchers to design a proper comparison approach. To the best of our knowledge, this work is the most relevant in the area [38,39]. The proposed framework consists of the following phases (Figure 3):

### 2.1. Reason for Comparison

The phase “Reason for Comparison” requires a clear definition for the reason that motivates the comparison process. The reason can be related, for instance, with an evaluation between different versions/implementations of the same algorithm, the best choice of algorithms for a specific scenario (e.g., real work problem) or comparisons between a classic algorithm with an innovative algorithm.

Besides the reason(s), it is also important to clarify which aspects are more important for the algorithms. For instance, should the algorithm adapt/present a solution for general situations or only for specific situations (e.g., it should include only linear problems or also consider convex universes?) What is the running time limit of each algorithm?

The reason and the relevant aspects play a critical role in the next phase definitions, which underline the need for a clear and weighted definition on the first phase.

### 2.2. Test Set

Considering the reason for the comparison, a test scenario must be defined, and to perform a correct assessment, all algorithms submitted to comparison should be applied on the same base. The test scenario may have three origins: real-world, pre-generated, or randomly generated scenarios.

Pre-generated and randomly generated datasets are often used to compare evolutions and/or new approaches to a specific reference problem, providing information about the algorithm’s features and behaviours (e.g., datasets used on scientific challenges). Random approaches results can become difficult to relate to real use case problems. Real-world datasets provide specialised information on real use case problems but can present as a setback the sample size limitation or the limited access to data (e.g., non-public datasets due to privacy reasons).

When defining the test scenario, it is important to consider the number and diversity of problems to embody, or the possibility that the dataset contains biased starting points. For instance, a small number of problems or a very specific problem can present a higher performance with a specific algorithm, influencing the overall view over other algorithms, resulting in an inadequate and biased analysis. A good example of this problem is an algorithm that performs better on small numbers of features or in compact search spaces.

### 2.3. Perform the Experiments

The experiments’ performance depends largely on environmental factors, namely the characteristics where experiments were conducted. The most obvious factors are the hardware and software (e.g., CPU or GPU capacity, or the operating system) but the programmer’s skills, programmatic framework, or code compilers are also non-negligible. The second relevant factor is the algorithm itself, which is exactly what the comparison seeks to assess.

To proceed with an evaluation, it is necessary to define the evaluation metrics, which will be collected during the algorithm’s execution on the test set. The measures must be defined according to the first phase definition and may be classified into three categories: efficiency, reliability, and solution quality.

Efficiency refers to the consumption of resources that the algorithm needs to run successfully. For instance, the algorithm’s total running time or the number of cycles needed to produce a solution. Reliability is related to the probability of achieving good performance on a variety of problems and can be evaluated, for instance, on the dispersion of the presented solutions for non-deterministic problems or by the success rate on deterministic situations. Finally, solution quality is evaluated considering two possible variants. In a known solution, a fixed-target variant is adopted, which evaluates the necessary time to reach the known solution; in the variant unknown solution, a fixed-cost approach is defined, which executes the algorithm according to a defined cost, e.g., time, and evaluates the error/difference between the current result and the best-known solution. For unknown solutions, which is quite frequent in real-world scenarios, it may be applied a fixed-cost variant, considering the best-known solution, or by estimating the optimal solution using statistical techniques.

### 2.4. Analyse and Report the Results

When analysing the results, the use of graphics and tables is usually used to assess the results obtained from the measurements and comparisons. These are powerful and versatile tools, which can improve human readability and the exploration of both generic and detailed perspectives.

Numerical tables can describe in detail the experiments measures, which offer a certain level of completeness, but it can also be overly presented, making it harder to extract useful information. Summary tables are often a good solution since they are easier to read, allowing quick insights about the data. These tables can be enriched with qualitative measures that summarise numerical metrics (e.g., run time qualified as acceptable or excessive). However, a summary comes with a cost, namely the loss of some detailed information that may be relevant.

Graphics, as visual methods, have the potential of providing better and more complete information to the reader, but for achieving the goal, it is recommended to have skills in the area or at least develop the charts considering graphical principles and best practices. A poorly drawn chart can easily confuse or mislead the reader, leading to wrong conclusions. Graphics can take diverse forms, from the simplest histogram or scatter plot to a more complex box plot or convergence plot, or even more advanced charts with regions, curves, and intervals.

## 3. D CNN NAS Testing and Measurements

The comparison experiments were conducted under the methodology described in the previous chapter, assuring a careful, non-biased and comprehensive evaluation. This chapter describes in detail all phases and decisions of the process.

### 3.1. Reason for Comparison

When dealing with sensors, regardless of the application area, it is common to find data on the 1D shape. For instance, in the healthcare area, one of the most common diagnostic methods is the electrocardiogram (ECG), which produces 1D data. While developing research on the medical area involving ECG medical exams, it was necessary to develop a 1D CNN able to process the ECG waves for a real classification problem (a supervised machine learning problem), supported by real patient data.

The performance measured by the accuracy of the network was crucial for the success of the research, and the need for defining the optimal 1D CNN architecture for the problem arose. After analysing related works, it was possible to understand that the problem could not be solved with a mathematical formulation or other similar approaches and, thus, it was required to perform an optimal hyperparameter combination search. The concept of optimal hyperparameter combination search for neural networks is named Neural Architecture Search, commonly known as NAS.

The literature review also revealed that NAS could be developed following some alternative strategies, but no comparisons between alternatives were found and it was not possible to understand which options would lead to the best solution, namely find the optimal hyperparameter combination for the 1D CNN for a given problem. It is also important to highlight that, even though it was possible to find some work related to the 2D CNN optimisation architecture search, it was not possible to find NAS approaches for 1D CNNs.

The motivation for creating this comparison was to understand which NAS approach would be able to find the best model architecture, or technically, the set of hyperparameters which perform best for a given problem. The ECG classification was the use case chosen as input to support the comparison. Furthermore, the motivation can be broken down into the following aspects (by order of importance):Model architecture with the best accuracy: the performance of neural networks can be measured through several different metrics and according to the machine learning problem typology. Some examples of metrics are root mean squared error, mean squared error, or adjusted *r* squared (R^2^) for regression problems, or the accuracy, precision, or F-Score for classification problems. In the present use case, a classification problem, the best metric is the classification accuracy.Algorithm execution time: the critical success factor was the 1D CNN accuracy, and there was no need to define the neural network architecture in real-time or in a very short period. Still, it was a requirement to find the optimal architecture on a realistic timeframe, which, has been defined, according to the team requirements, as “good” if the total running time is below 24 h and “excessive” if the total running time is above 32 h, considering the problem complexity. Twenty-four hours usually corresponds to a working day. It is considered as “excessive” (above 32 h) since it requires waiting for the results for more than two working days.Resource’s consumption: the democratisation of machine learning tools and methods is only possible if they can be executed on accessible/affordable computational architectures. In this study, it is considered that the NAS should be executed on a relatively cost-accessible infrastructure for any research centre in most of the countries in the world.

### 3.2. Test Set

As previously referred, the need to build an optimised model based on 1D CNN arose from a real-world use case research scenario, where there was the need of classifying ECG waves (a supervised machine learning problem). The dataset used as the test set corresponds to the real-world use case dataset.

The ECG data was collected by the Generation XXI project, a previously established prospective population-based birth cohort from Porto, Portugal, that aimed to characterise prenatal and postnatal development throughout childhood, adolescence, and adulthood. This project followed up a total of 8647 children born between April 2005 and August 2006 and gathered several ECGs among the cohort population at the 10 years’ evaluation.

The data was extracted from an ECG equipment that was assigned to the Generation XXI project, in a proprietary vendor XML format. Data was processed by the research team, which has performed several machine learning tasks to prepare and achieve a final dataset.

The preparation phase has included a data understanding phase, where the XML detail information was analysed, and relevant data (future features) were identified. The following information was extracted from the XML format: ECG collection date; individual related data, namely the identification (encoded), birthdate, weight, height, gender, ECG signal (Lead I), and ECG report. Regarding the ECG signal, the records presented a sampling rate of 500 samples/s, with a ten-second duration, which results in a total of 5000 measures for each record. The ECG was collected in a raw format, which means noise filtering was not performed. The data, which included records from 5646 children, was also inspected to find duplicates, records with missing and/or abnormal signal data, incorrect birth date or collection time, invalid weight and/or height values. The data understanding phase was completed with an extensive statistical and data analysis stage.

The data preparation phase included the creation of derivative features, elimination of records that presented cardiac abnormalities (identified by the ECG report feature in the form of text), the ECG signal filtering with a Finite Impulse Response (FIR) and a band-pass filter, and the decomposing of each of the 10,000-millisecond signals in beats, supported by the identification of R-peaks followed by the beat extraction.

The process resulted in the extraction of a total of 14,308 complete beats, with a 600 ms duration, composed by 300 measures (one measure by each 2 ms).

Figure 4 presents one of the beat records which comprises the final data composed of 14,308 records.

All performed tests were based on the same testing set, exactly on the same conditions, following the same training process, including record order.

### 3.3. Perform the Experiments

As discussed in the methodology section, the experiments rely significantly on the environmental factors, which include computational settings, but also the programmer’s skills and the programmatic environment.

In order to minimise noise or distortion on the experiments, the same machine with the same settings were used. The research team also “froze” any software update that could be performed automatically, turned-off schedule works and limited the access to the environment only for the evaluation purposes. To avoid any influence of the programmer’s skills in the NAS processes and strategies, as well as the implementation of the comparison, only a single programmer worked on the entire development. The CNN’s model training was performed exactly in the same way, based on the Keras framework, assuring the focus on the NAS search strategies comparison, and how strategies combine the hypermeters. A highlight for the fact that the combinations of training hyperparameters influence in fact the training. Finally, the same implementation language was used, in this case, Python 3.7.

The computational infrastructure that supported the experiments had the following settings:Operating System: Ubuntu 18.04Disk: 500 GB SSDSystem RAM Memory: 32 GBCPUs: AMD Ryzen 73,600XGPUs: NVIDIA GTX 1080 TI 11 GB + RTX 2060 SUPER 8 GB

This configuration was chosen under the principles defined on the phase “Reason for the Comparison”. This configuration, for being a simple hardware set, is also a very affordable infrastructure with an interesting computational capacity.

For the experiments, considering the relevant algorithm metrics to be assessed, the following measures were monitored:Model accuracy: In this current use case, the research team has defined classification accuracy as the suitable performance metric. The accuracy was measured in the three sub-datasets extracted from the initial dataset. The sub-datasets were named as training, validation, and testing sets, and the split from the original set was 70%, 20%, and 10% for each set, respectively. The goal was to achieve a model that best generalises the learning on data and, thus, a solution where accuracy values were homogeneous and consistent was considered as a good solution, instead of “simply” the best accuracy values of any of the sub-datasets.Running time: as previously defined, there was a limit (defined by the research team) for the execution time capped by 32 h and a desirable run time lower than 24 h. For that reason, the total running time was measured.Resource’s consumption: the benchmark experiments and NAS strategies usually consume a considerable amount of CPU and GPU resources, especially GPU usage and memory. So, it was necessary to be constantly monitoring those metrics.

In summary, the experiments were measured on the quality of the solution (variant unknown solution) on the accuracy of the three partial data sets: and on the efficiency through execution time and resource consumption.

As described in previous sections, it was not possible to find NAS frameworks or tools that processed 1D CNNs and those were necessary to develop/adapt/extend new processes/methods for 1D CNN NAS. The NAS strategies (depicted in Figure 5) considered in the current comparison followed two main approaches, namely:Approach A: Custom processes based on AutoKeras, an AutoML library that provides some tools for hyperparameter optimisation. The developments have included four different strategies: greedy, random, Bayesian, and hyperband [40,41].Approach B: a new development based on the Deep Neural Network Evolution (DEvol) library, a basic proof of concept for genetic architecture search in imaging data, made available to the community [42].

The experimental testing design was the following:Twenty-five experiments for each scenarioScenarios:
○Approach A
▪A1: Greedy▪A2: Bayesian▪A3: Hyperband▪A4: Random○Approach B: Genetic

The greedy approach is an algorithm that can find the best solution in a limited search space because it tries out all possible combinations. In a non-limited space or where not all possible solutions can be tested, the algorithm tries the possible combinations within a defined timeframe or number of attempts. The search strategy consists of trying all possible values for each variable and then changing only one in each iteration. The random approach is similar, but in this case all variables are chosen randomly in each iteration. Although trial-and-error, random and greedy strategies are known to be the most common approaches, in more recent 1D CNN research (as previously commented in the introduction section), there is no orientation/direction in the search process to find the best solution.

Bayesian search considers the last combination of variables and the corresponding score/result when defining the parameters for the next iteration. The algorithm optimises the iteration possibilities, i.e., it tries to search in promising areas of the search space, increasing the probability of achieving the best combination of variables.

The concept of the hyperband search consists of eliminating or early stopping the solutions that do not perform well and those that do not “seem promising”, to reduce computational resources. It starts with a small exploitation in each of the possible combinations, and if the performance of this attempt does not perform well, the combination is discarded for the next iteration of the algorithm. This cycle repeats itself until there is only one combination left.

The genetic algorithm is a search algorithm inspired by the theory of natural evolution. The concepts reflect the process of natural selection where the best individuals in a population are selected for reproduction, and the next generation is expected to inherit the best characteristics of the best parents, contributing to an improved next generation. In this particular case, the characteristics of the individual are the hyperparameters choices/combinations, specifically the ones described in the next paragraphs. It is an iterative process that seeks to achieve a final population made up of the fittest individuals, which hopefully includes the best possible solution for a given problem.

For a genetic approach, five phases are considered: the initial population; fitness function; selection; crossover; and mutation. The initial population is usually randomly defined, and each characteristic (also known as genes) is joined together, creating an individual (possible solution). The fitness function defines how individuals are evaluated assigning a fitness score, which in this case is the performance obtained by the individual (combination of hyperparameters) training process. The selection is the process of selecting the best individuals that will be used for reproduction, creating the next generation. Usually, only 10 to 20% best individuals are selected. The crossover consists of choosing a random point of the genes and performing the exchanging of genes among the parents. That process is illustrated in Figure 5. where the crossover was performed on the third gene and the best parents conceived two new children. Finally, the mutation occurs when some of the genes of the new individuals suffer a change based on a low random probability. The algorithm termination can be defined based on the population convergence, meaning the new generations are not able to present significant changes when compared with previous generations, or limited by a number of generations.

All the previous scenarios perform a search for the best CNN hyperparameters, including the structural and training hyperparameters, following each of the described strategies. The considered structural hyperparameters were the number of layers (convolutional, pooling, and fully connected layers) that compose the CNN structure, as well as the number of filters, kernel sizes, padding, stride, activation functions, normalisation techniques, dropout percentages in each layer and number of neurons (for the fully connected layer). Regarding the training parameters, the search was performed on the network optimisation function, learning rate, batch size, weights initialisation, bias initialisation, and momentum.

For a fair comparison between approaches and considering the NAS goals, the search settings were defined in a way that guaranteed equal opportunities for all scenarios. NAS configuration settings: epochs: 200; max trials: 400; generations number: 20; population size: 20.

The application of the five scenarios and the corresponding experiments have resulted in the best 1D CNN architecture for the problem (on the study outcomes), which is presented in Figure 6.

The neural network structure presented the following configuration:Input layer: 300 inputs to which correspond the ECG signal measuresConvolutional block 1:
○Composed by 128 filters; 1-D kernel with size 3 and stride = 1; ReLU activation function and a dropout of 0.45Convolutional block 2:
○Composed by 16 filters; 1-D kernel with size 3 and stride = 1; Sigmoid activation function and a dropout of 0.1Convolutional block 3:
○Composed by 8 filters; 1-D kernel with size 3 and stride = 1; batch normalisation layer; ReLU activation function and a dropout of 0.05Convolutional block 4:
○Composed by 128 filters; 1-D kernel with size 3 and stride = 1; Sigmoid activation function and a dropout of 0.4Flatten layer: 38,400 neuronsFully connected/dense block: 32 neuronsOutput: Classification of ECG waves in three classes, namely “normal/control group”, “overweight”, and “obese”. By inspecting ECG waves, it was possible to “blindfold” understand, which was the individual body mass index classification. Note: This classification served other research goals related to the existence of cardiac changes and their identification.This classification was achieved based on three neurons (one-hot encoding), where each of the neurons represented a class. A SoftMax activation function was used to ensure that the output values were between 0 and 1. The sum is 1, which represents the probability of belonging to each class.

The training hyperparameters were the following:Optimisation function: ADAM OptimiserLearning rate: 10^−3^Batch size: 200Early stopping criteria: 30 optimisation steps based on the validation setLoss function: categorical cross-entropyWeight’s initialisation: Xavier initialiserBias initialisation: zero

### 3.4. Analyse and Report

Considering the reason for the comparison and the important aspects to evaluate, it was decided to create a summary metric table with the accuracy outputs for each scenario from the 25 experiments.

To evaluate the results achieved by the models, the table contains the accuracy measured in the three different partial data sets, namely training, validation, and testing, as well as the most consistent model. The training set was used to minimise an error function in the modelling phase. The results and the quality of the solution are measured by analysing the performance of the validation and test subsets. The most consistent model is obtained by analysing the 25 experiments for each scenario and measuring them from the difference resulting from the accuracy of the validation and test datasets in each iteration. A stable model shows better generalisation resulting in more consistent results, which means less difference between the two accuracy values.

Considering the non-deterministic nature of a 1D CNN implementation and the search strategies applied on the NAS, the research team has considered the variability/dispersion in the performance results to be relevant, and, thus, box plots were developed to analyse output stability. This dispersion analysis allows avoiding a noisy or distorted evaluation due to a “lucky shot” on any of the scenarios.

A summary table, the most consistent model, and the accuracy box plot are presented in Table 1, Table 2, Table 3, Table 4 and Table 5 and Figure 7, Figure 8, Figure 9, Figure 10 and Figure 11, for each of the scenarios. The presented accuracy is based on the classification performance (multi-class accuracy), which is defined as the average number of correct predictions. For this measurement, a confusion matrix was used since it presents both predicted and actual/true class summary, highlighting the sum of true and false predictions.

The second measure defined as relevant for monitoring was the execution time. The execution time was obtained by measuring the execution time of the algorithm from the initial start of the algorithm to the result consisting of the best network found. Recall for the NAS settings, namely 200 epochs, 400 max trials, 20 generations, and a population size of 20 samples. 

Table 6, Table 7, Table 8, Table 9 and Table 10 shows a summary of the execution times for each scenario.

The current comparison process includes the monitoring of computational resource consumption. In Figure 12, it is possible to find CPU and GPU resource consumption, including GPU memory, for each scenario. The resource consumption was analysed for each scenario considering all 25 experiments. By analysing the CPU and GPU monitoring results on different experiments, similar results were obtained, so it was decided to provide a resource monitoring sample for each scenario. Based on the CPU consumption, the vertical axis in the figures shows the percentage of the CPU used.

Regarding GPU resource consumption, the test environment infrastructure included two GPUs. Figure 13 presents the percentage of GPU usage and GPU memory usage.

The plots were created using the data generated by the experiments and managed by the open-source library ClearML—Auto-Magical Suite of tools to streamline your Machine Learning workflow Experiment Manager, ML-Ops and Data-Management [43].

Considering the experimental tests and the accuracy metric, it is possible to verify that:The best accuracy in the validation sub-dataset was achieved by Approach A1 and Approach A4 achieved a value of 0.8106 (%) in the testing sub-dataset.By looking more closely at the full tables of results, it is possible to verify that the value obtained on Approach A4 (random) for the test sub-dataset was a “lucky shot”, and when observing the distribution of results through the box plot, we confirm that this value is clearly an outlier.Approach A3 (hyperband) presents the lowest values for the worst outcomes of all subsets of data. It also presents the lowest scores when considering the mean results. At first glance, this approach seems to have the worst accuracy ratings, but in order to understand these values and to draw a conclusion, it is important to understand how the hyperband approach behaves. We recall the description of the hyperband strategy (on a previous section), which relies on an early stopping strategy for resource allocation and run time optimisation.The approach that achieves the best mean accuracy is Approach B on the test sub-dataset and Approach A1 on validation set.Regarding standard deviation, Approach A3 presents the highest values for the validation and test sub-dataset. The lowest values are presented by A4 and B, respectively, for validation and testing sets.Analysing the boxplots (Figure 7, Figure 8, Figure 9, Figure 10 and Figure 11), Approach A3 stands out as the one presenting the most narrows interquartile range, and the one with the most outliers. Both approaches A1 and B show consistency between the three partial data sets, which show box plot shapes with similar accuracy. Two differences stand out among these approaches: (i) the three subsets of data have similar shapes, but the training, and mainly the testing have higher scores in approach B; (ii) in the test subset, A1 represents higher values as outliers, with B representing outliers at the lowest values. A2 is the approach with the lowest consistency, as well as the smallest values on the accuracy distribution in the testing box plot.The results of the box plot analysis are confirmed by comparing the “most consistent model” for each approach. Approach B shows the smallest deviation in the validation and test subsets with accuracies of 0.7960/0.7980 (%), followed by Approach A1 with 0.7934/0.7903 (%). The highest deviation is presented by A2, followed closely by A4.

Considering the running time analysis, it is possible to verify:With an execution time (at least) eight times shorter, the A3 approach performs better than all other variants. This can easily be explained by the approach behaviour (hyperband), previously described.Approaches A1, A2, and B showed similar values. These results were expected because all approaches perform experiments similarly. The biggest difference lies in the search design, whereby A1 is “dummy”, A2 offers an “oriented” search, and B defines new possible solutions based on genetic algorithms.Scenario A4 shows a reduced variation considering the total running time.All approaches carried out in an acceptable time, which, given the definition of the study, was 32 h. Most were able to run under the desirable 24 h timeframe.

Regarding the computational resource consumption:In all cases, the CPU usage was below 22% and the system memory usage was below 16%. (Available RAM: 32 GB). It is also important to note that the experiments were configured to use GPUs for CNN processing to optimise convolution operations. Hence, it was expected that it would not have high CPU or system memory consumption.All approaches used the same training process that uses all the available GPU memory. For this reason, the approaches have a similar memory consumption (note that some of the GPU-1 memory is reserved for system tasks). The exception can be found in the A3 scenario, which starts with a few epochs in every training session and then reduces the number of trained models.The utilisation of both GPUs is identical in all scenarios, showing a parallel consumption. This behaviour was expected because the training was configured to distribute processing among the GPU units.The GPU utilisation ranges from 10 to 90% in approaches A2, A4, and B and is significantly lower in A1 and A3.

## 4. Conclusions and Future Work

This manuscript begins with a brief overview of deep learning algorithms and more specifically on CNNs, including 1D, 2D, and 3D convolution processing alternatives. The article characterises the problem (the research question) that consists in optimising a 1D CNN architecture (best options for training and structural hyperparameters) and introduces the concept of NAS as a solution. To optimally achieve the goal of optimising a 1D CNN, and since the existing literature does not provide, for now, an optimal way for reaching an optimised 1D CNN (the research gap), five different search approaches/procedures have been implemented, tested, and measured, as part of the study. The developments were shared with the community in order to help the academic and scientific community, and especially researchers using 1D CNN. Conclusions of the best NAS approach to 1D CNN problems and suggestions for possible future work were drawn and presented.

The study was developed based on a real-world use case that involved real sensor-based medical diagnostic methods, namely ECGs, and the challenge consisted in implementing a well-performant classifier for 1D data.

The study developed, compared, and analysed five NAS strategies, namely greedy, random, Bayesian, hyperband, and genetic, aided by the method presented by Vahid Beiranv et al. in the article “Best Practices for Comparing Optimisation Algorithms”. For the experiments and analysis, several dimensions were considered, namely accuracy (with different perspectives), total execution time and resources consumption.

Accuracy was measured by the best results on the validation and testing sets, as well as the models’ consistency and dispersion of the results. Though approach A1 (greedy) has reached good results on the accuracy, approach B (genetic) has the best outcomes when analysing the model’s consistency and the dispersion of the results (best observed on the presented box plots).

Regarding efficiency, measured by the computer system resources usage and total execution time, all algorithms had a very acceptable behaviour with decreased consumptions of CPU and system memory. Regarding GPUs, the algorithms have taken full advantage of their capacities for CNN training, with high use of the available resources. Approaches A1 and A3 were a little less demanding. This outcome was expected considering that one of the 1D CNN advantages is a reasonable demand of computational resources.

Total running time was acceptable in all approaches. The average run time was below 24 h, which the study defined as a good total execution time. Even the worst cases exceed this limit on a maximum of 7 h, which was still acceptable. A final note for the approach A3, which stands out with the need of only 1 h processing, which is 10 times less than any other. This approach is a good choice when facing a short time limitation and a less performant network is acceptable.

Considering the study definitions and purpose, the research team concluded that the NAS strategy that best suits the requirements was the approach B, the genetic approach, which uses affordable resources (like the other alternatives) and has produced the most performant architecture for the present use case challenge.

The optimisation of neural networks in general, and mainly for 1D CNNs, is critical for the extraction of data’s true value and meaning. Nevertheless, the literature review showed that, even in the most recent works, the common approaches fall back to trial-and-error methods and grid searches with limited parameters. These current approaches present a drawback to non-effective optimisation of networks, meaning potential wasted data value. The NAS approach and the resources presented, intended to contribute to the model’s optimisation, and consequently to improved research results. The use of NAS presents the disadvantage of adding one more phase to the already complex machine learning pipeline. In our opinion, the benefits greatly outweigh the costs.

For future work, our suggestions/proposals are:

The current study scope is focused on 1D CNNs, which, as previously mentioned, are focused on 1D convolutions. CNNs, besides 2D convulsions that already have the attention of the research community, also provide 3D convolutions which, like 1D, presents huge potential. It may be interesting to extend/develop the current NAS scenarios to 3D CNNs.

The current work was supported by a real-world use case including several instances of a medical diagnostic method based on sensors, namely electrocardiogram waves. One dimensional data, namely when the signal/measures sequence are relevant, can benefit from 1D convolution processing. This work was able to achieve success with interesting performance model architectures, in this case, measured by accuracy, but the study is focused on a single dataset, which is a study limitation. We believe that there is a huge potential to apply the study outcomes in other 1D domains, including challenges with regression outputs. The possibility of using other datasets and/or other types of model outputs will enrich the knowledge about 1D CNN and NAS applied on 1D CNN.

Two dimensional CNNs, and other image processing alternatives, have been quite explored by the scientific community and it is easy to find 2D reference datasets that support neural network studies, comparisons, and research, as for instance CIFAR-10, CIFAR-100 (colour images presenting 10 classes such as airplane, cat, or truck) or MNIST (handwritten digits). Public 1D datasets are scarce, and in our opinion, it is important to create 1D reference datasets to progress on the 1D machine learning area. Furthermore, we would like to contribute to these repositories.

The developments performed in the study scope were shared with the community to enhance the evolution of the 1D CNN area and as a resource for 1D CNN researchers.

This work has been able to reach a performant CNN architecture, using the genetic NAS approach, which resulted in a solution for the real use case, a 1D classification problem (the origin of the research). The goal was achieved with affordable computational resources and in a quite acceptable timeframe. Because it is unknown information, it is not possible to claim that the most optimal architecture was found, but it is possible to claim that the best-known architecture was achieved. Based upon the results, and the fact that the current problem has an unknown solution, the study was deemed a success.

The research team is confident that the work will provide valuable knowledge and resources to future 1D CNN researchers in the domain of sensors and other 1D data sources.

## Figures and Tables

**Figure 1 sensors-21-07990-f001:**
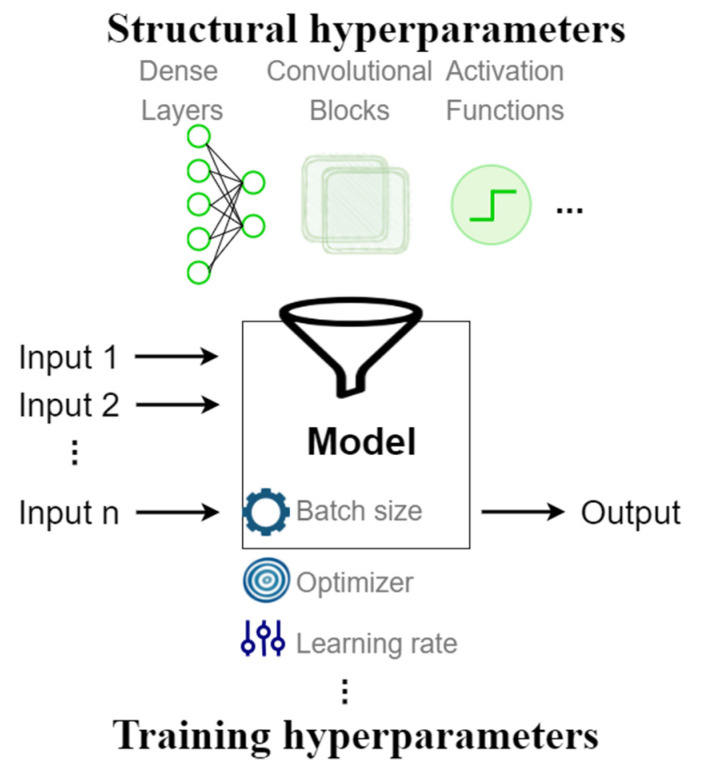
An example of a CNN architecture definition (including hyperparameters).

**Figure 2 sensors-21-07990-f002:**
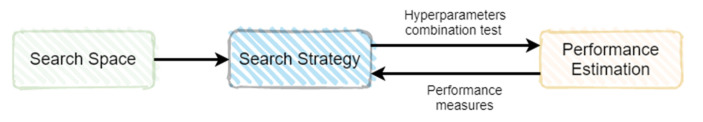
Basic concept of Neural Architecture Search (NAS).

**Figure 3 sensors-21-07990-f003:**
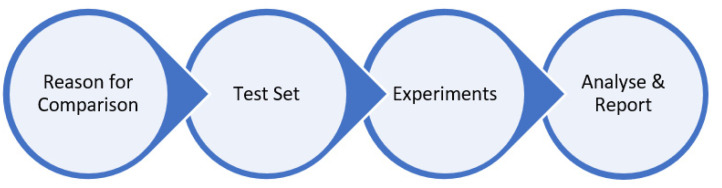
Comparing methodology phases.

**Figure 4 sensors-21-07990-f004:**
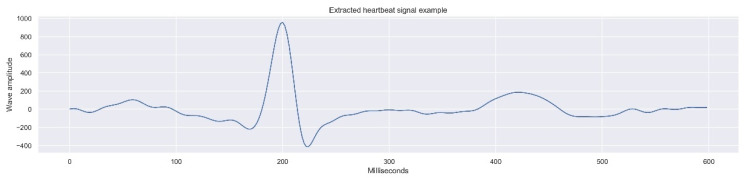
One dimensional data example (heartbeat signal), part of the final dataset.

**Figure 5 sensors-21-07990-f005:**
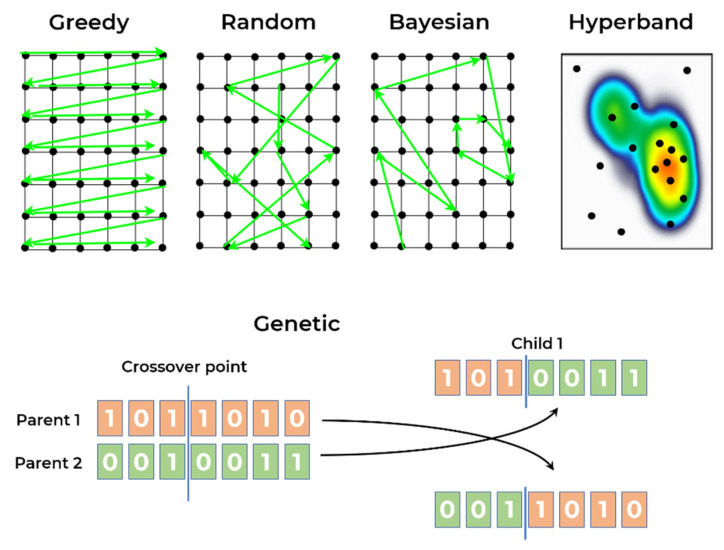
Illustration of the NAS strategies applied in experiments.

**Figure 6 sensors-21-07990-f006:**
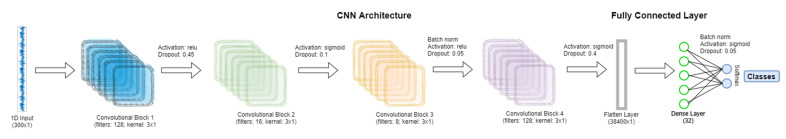
Best 1D CNN architecture produced by the NAS process.

**Figure 7 sensors-21-07990-f007:**
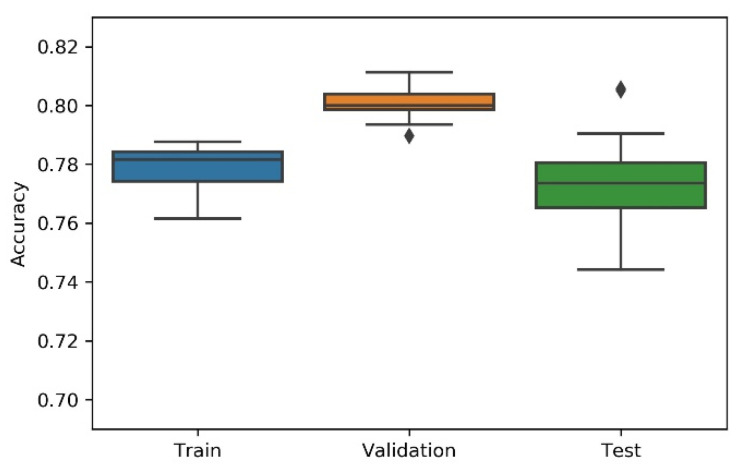
Experiment’s performance for Approach A1 (Greedy) accuracy (in %).

**Figure 8 sensors-21-07990-f008:**
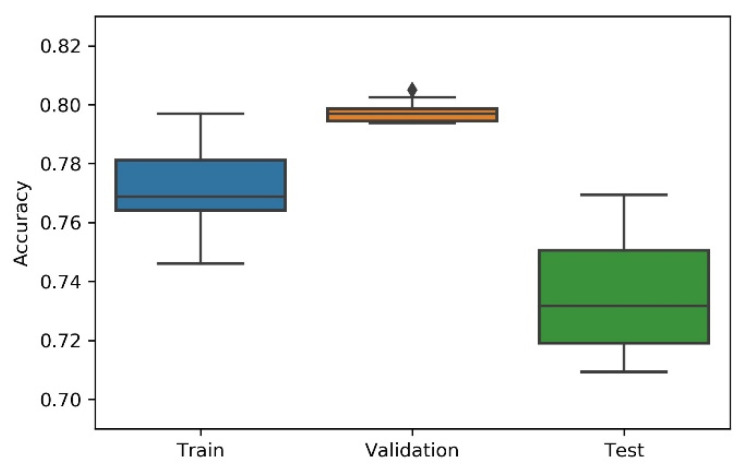
Experiment’s performance for Approach A2 (Bayesian) accuracy (in %).

**Figure 9 sensors-21-07990-f009:**
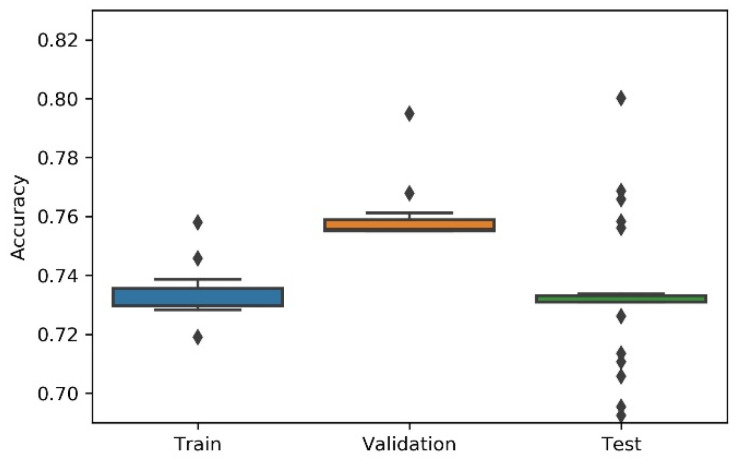
Experiment’s performance for Approach A3 (Hyperband) accuracy (in %).

**Figure 10 sensors-21-07990-f010:**
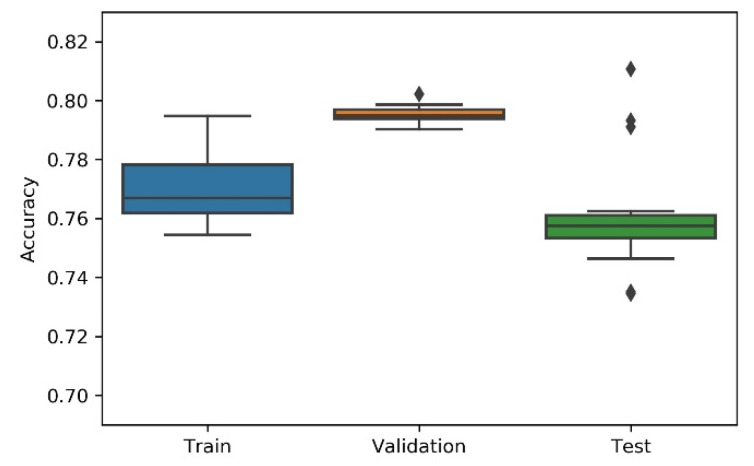
Experiment’s performance for Approach A4 (Random) accuracy (in %).

**Figure 11 sensors-21-07990-f011:**
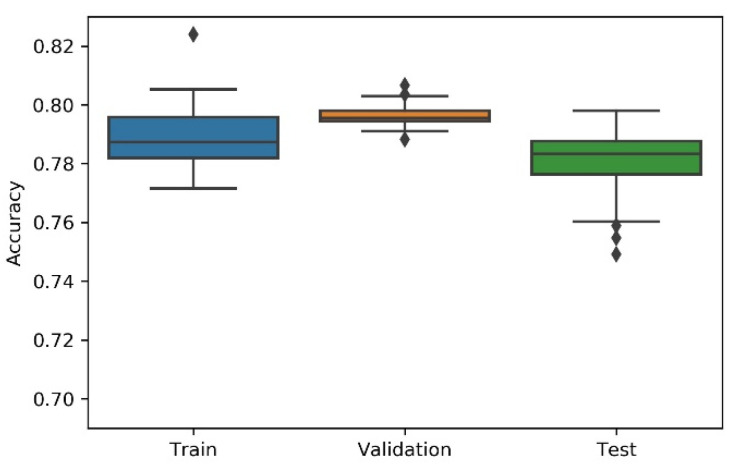
Experiment’s performance for Approach B (Genetic) accuracy (in %).

**Figure 12 sensors-21-07990-f012:**
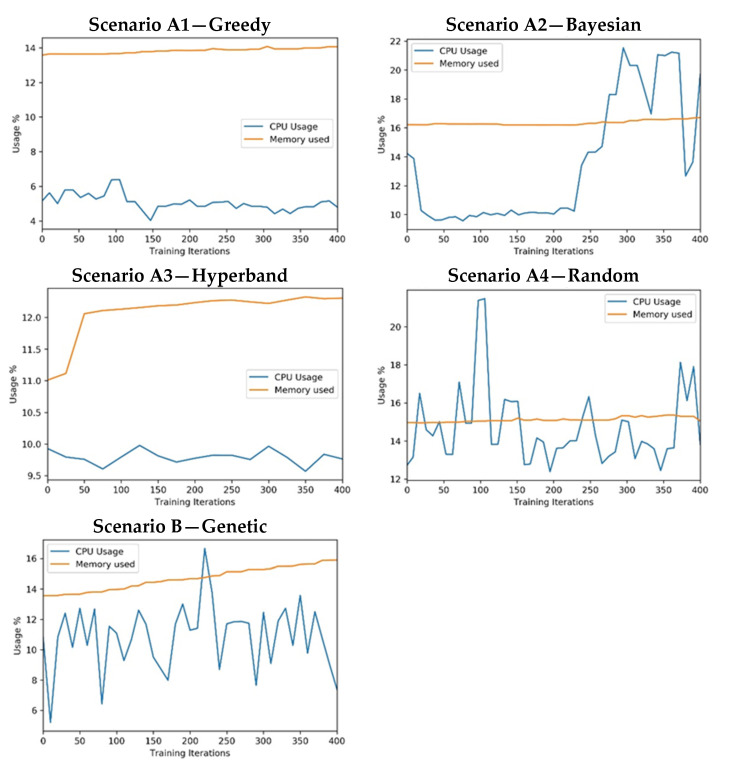
Experiment’s measure: CPU consumption.

**Figure 13 sensors-21-07990-f013:**
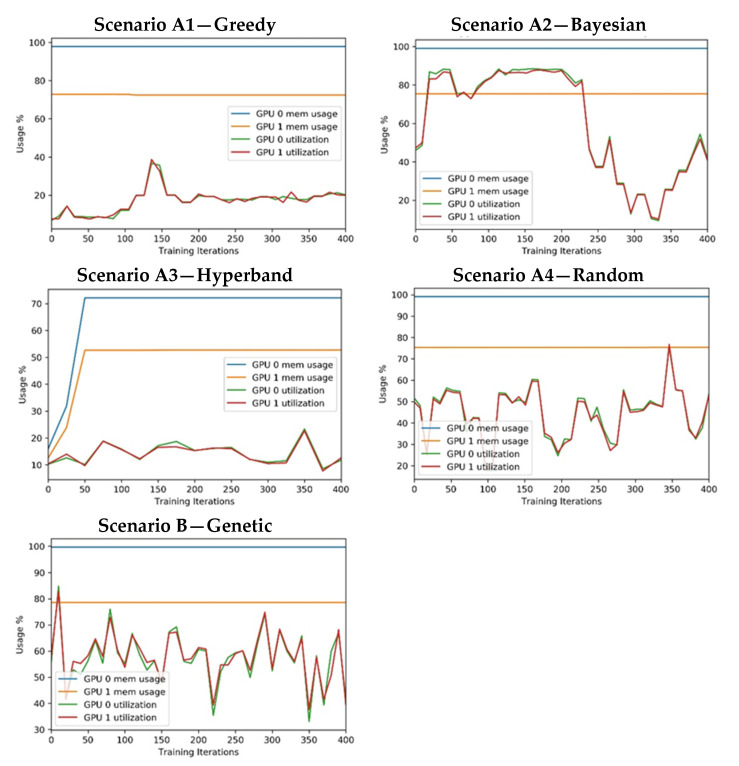
Experiment’s measure: GPU consumptions.

**Table 1 sensors-21-07990-t001:** Summary table for Approach A1—Greedy accuracy (in %).

Accuracy in %	Train (%)	Validation (%)	Test (%)
Best	0.7876	0.8113	0.8057
Worst	0.7616	0.7896	0.7442
Mean	0.7784	0.8007	0.7737
Std dev	0.0070	0.0049	0.0144
Most consistent model		0.7934	0.7903

Approach A1—Greedy.

**Table 2 sensors-21-07990-t002:** Summary table for Approach A2—Bayesian accuracy (in %).

Accuracy in %	Train (%)	Validation (%)	Test (%)
Best	0.7969	0.8050	0.7694
Worst	0.7460	0.7938	0.7093
Mean	0.7711	0.7971	0.7353
Std dev	0.0150	0.0029	0.0178
Most consistent model		0.7945	0.7624

Approach A2—Bayesian.

**Table 3 sensors-21-07990-t003:** Summary table for Approach A3—Hyperband accuracy (in %).

Accuracy in %	Train (%)	Validation (%)	Test (%)
Best	0.7580	0.7948	0.8001
Worst	0.7190	0.7550	0.6925
Mean	0.7328	0.7587	0.7332
Std dev	0.0072	0.0081	0.0231
Most consistent model		0.7613	0.7582

Approach A3—Hyperband.

**Table 4 sensors-21-07990-t004:** Summary table for Approach A4—Random accuracy (in %).

Accuracy in %	Train (%)	Validation (%)	Test (%)
Best	0.7948	0.8022	0.8106
Worst	0.7544	0.7903	0.7345
Mean	0.7709	0.7955	0.7597
Std dev	0.0127	0.0025	0.0166
Most consistent model		0.7983	0.7932

Approach A4—Random.

**Table 5 sensors-21-07990-t005:** Summary table for Approach B—Genetic accuracy (in %).

Accuracy in %	Train (%)	Validation (%)	Test (%)
Best	0.8240	0.8067	0.7980
Worst	0.7716	0.7882	0.7491
Mean	0.7887	0.7964	0.7799
Std dev	0.0116	0.0040	0.0124
Most consistent model		0.7960	0.7980

Approach B—Genetic.

**Table 6 sensors-21-07990-t006:** Experiments measures: running time for Scenario A1—Greedy.

Run Time	Time (h:m:s)
Best	08:08:14
Worst	30:55:25
Mean	13:48:27
Std dev	04:56:32

**Table 7 sensors-21-07990-t007:** Experiments measures: running time for Scenario Scenario A2—Bayesian.

Run Time	Time (h:m:s)
Best	08:25:00
Worst	28:51:00
Mean	16:55:57
Std dev	04:55:31

**Table 8 sensors-21-07990-t008:** Experiments measures: running time for Scenario Scenario A3—Hyperband.

Run Time	Time (h:m:s)
Best	00:57:00
Worst	1:05:00
Mean	01:01:07
Std dev	00:02:27

**Table 9 sensors-21-07990-t009:** Experiments measures: running time for Scenario Scenario A4—Random.

Run Time	Time (h:m:s)
Best	11:19:00
Worst	12:13:00
Mean	11:37:04
Std dev	00:09:41

**Table 10 sensors-21-07990-t010:** Experiments measures: running time for Scenario Scenario B—Genetic.

Run Time	Time (h:m:s)
Best	09:21:00
Worst	28:20:00
Mean	14:58:40
Std dev	04:46:36

## Data Availability

Not applicable.

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
