# Peer review of "Neural Architecture Search for 1D CNNs—Different Approaches Tests and Measurements"

_sensors, 2021, doi:10.3390/s21237990_

Round 1
Reviewer 1 Report
- Does the paper demonstrate an adequate understanding of relevant literature in the field and cite an appropriate range of literature sources? It seems that a few related works are included in the paper. More recently published papers should be discussed in related work section, and also be compared to the proposed method.
- Is the paper's argument built on an appropriate base of theory, concepts or other ideas? Has the research or equivalent intellectual work on which the paper is based been well designed? How authors justify that the methods employed are appropriate?
- The paper does not identify clearly any implications for research, practice and /or society? How does the paper bridge the gap between theory and practice? How can the research be used in practice (technical and resource impact), in teaching, to influence public policy, in research (contributing to the body of knowledge)? What is the impact upon society (influence public attitude, affecting quality of life)? Are these implications consistent with the findings and conclusion of the paper?
- How data is created and using what parameters simulations were conducted? How authors can say that it is emulation and resembles real world data?
- Authors have not presented limitations of this work. How this work can be extended in future?
- The experimental results (table-1) how to determine? What about the data selection and simulation parameters?
- Related work also needs restructured. Can you make a few categories instead of listing them as one very long paragraph? What are the most related work? And in association with the experiment, why you choose benchmarking with greedy and bayesian?
- Experimental results are good but needs more details? What is the need to evaluate proposed framework in this simulator only?
- The conclusion of the paper is very generic. I would again like authors to give a close attention to it and rewrite the part based on the experiment and related findings.Conclusion: No clear conclusion or outcomes why the proposed algorithm should be used over the existing ones.
- In the conclusion, the practical application field of the proposed methods and the research findings can be described that highlights the contribution of this article. Then, the advantages (and disadvantages?) of the proposed methods should be discussed.
- Have you analyzed the computational complexity of the greedy, random, Bayesian, hyperband and genetic approaches?
- Please define abbreviations before their usage. Furthermore, there are some typos errors in the text which leads to ambiguity while understanding the paper. The authors are advised to read the entire manuscript again to avoid such mistakes.
- After getting improved results over basic approach (which is ambiguity because it is not referenced) authors have not stated conclusion at a high note. Please add conclusion with more details and also show the future direction areas. Results methodology and comparative study should be reflected.
- Mathematical model must be presented in the form of equations by adding proper equation number.
- Simulations results presented in this paper are far from being comprehensive. The drawn conclusions must be supported by more simulations and additional scenarios must be analyzed.
Reviewer 2 Report
Summary:
This paper presents the approach of One Dimension Neural Architecture Search (1D NAS), which automates the search for the best combination of Neuronal Networks (NN) hyperparameters (model architecture), including both structural and training hyperparameters for 1D CNNs. The paper creates testing scenarios for five NAS strategies, namely greedy, random, Bayesian, hyperband and genetic approaches. Results are presented in terms of several metrics, including accuracy, including best accuracy, accuracy model consistency and accuracy variability, running time and computational resources consumption.
Comments:
- Overall, I find the paper is not easy to follow and my major concern is about the novelty of this work (evaluation of established methods).
- Very generic discussion of different approaches.
- I would recommend the authors to provide more training details for different NAS algorithms.
- English language and style must be improved.
Reviewer 3 Report
This paper focuses on One Dimensional Neural Networks (1D-CNN) and proposes an approach that automates the search for the best combination of hyperparameters that define the model architecture, including structural and training hyperparameters. The work
Concerns about the paper are the following:
- There are several paragraphs made of just one or two sentences in all sections. They should be grouped more cohesively.
- It is not clear which dataset was used for the experiments. The authors should describe it extensively.
- Authors should include more recent studies involving deep neural networks including 1D-CNN in different applied predictive tasks (see DOI: 10.3390/s21041235, 10.3390/s21196555, 10.3390/s21175936, 10.3390/rs12244142, 10.3390/s21165338).
- There is a typo in Figure 1: "Learning rare" : it should be "Learning rate"
- Figure 2: "NAS" is never used before and should be introduced in its extended form
- There is a typo in Figure 3: "Reason for comparison" : it should be "Reason for comparison"
- Figure 4: It is not clear what is reported on the x-axis and the y-axis
- Page 8: To improve readability, I suggest applying bold on each item keyword in the list of items.
- Figure 6-10 and Table 1: Tables should not be screenshots but actual tables in the manuscript according to the journal's template
I think that the paper provides an interesting overview of this model architecture which is useful, but the quality of the paper is uneven and rushed at this point.
I suggest the authors prepare a revised version of the manuscript.
Reviewer 4 Report
In this work a 1D CNN is optimized by means of five tools.
The author split the whole dataset into three subsets, training, validation and test, as it usually done when neural networks are used to carry out a certain task. The training dataset is used to train a model, validation data to select the model performing the best and, finally, the test set is used to prove how good the selected model is. Usually training and test datasets are merged into one only set which is used to search for the best model. So the performance of a classification tool, such as that used in this work, should be stated only with the validation and test dataset, as that obtained with the training data is used to train the model by minimizing an error function which is closely related to performance. In fact, errors used to train the training dataset may be different from those used for validation and test. The first is used to be minimized by a training algorithm while the second are used to measure the performance of the model tested. So, errors achieved with the trained dataset may not be used to state the good performance of a model nor to be compared with those obtained with the validation and test datasets, as it is done in section “Analyse and report”.
CNNs should be described in the text. The five optimization algorithms used should also be described. The structure of the 1D CNN model used in this work should be described in detail in the text. The hyperparameters to be optimized should be also described.
There is no point in justifying and describing why a classification model structures should be carried out. It is obvious that a classification model must be optimized before used. So, chapter 2 “Methodology” should be removed and substituted by a description of CNNs and the five optimization algorithms used.
Which are the input data? What does the model provide as output? How is the 1D CNN used in the work trained? What kind of training is used: supervised or unsupervised? Which is the structure of input and output data? If a supervised training was performed, which is the structure of input-output pairs? In other words, the structure and meaning of the data provided to the network and the information it gives as an answer (a prediction, a behavior pattern…) should be described in detail. In addition, the figures of merit of the CNN outputs (the difference between what the model provides and what it should provide) should be also described and provided.
What do the numbers in tables 6 to 10 represent? Are they errors (mean squared error, mean absolute error…)? What does “most consistent model” exactly mean? This should be clearly explained in the text and in the Figures.
In view of the above considerations, I think that this work may not be published in its present form. I think that the authors should have described the 1D CNN structure they propose, the parameters to be optimized, the task the CNN is devoted to and the structure performing the best.
Reviewer 5 Report
The manuscript proposes a one dimension convolutional neural networks addressing the concept of one dimension neural architecture search for automating the search for the best combination of neural networks hyperparameters.
As a major comment, authors are suggested to further discuss on the validity of the one dimension convolutional neural networks. The validity and the comparative analysis of the tests and measurements are to be discussed.
The manuscript includes several typos and grammatical errors.
Although the paper has appropriate length and informative content, several parts must be improved and written in better grammar and syntax. It would be essential if authors would consider revising the organization and composition of the manuscript, in terms of the definition/justification of the objectives, description of the method, the accomplishment of the objective, and results. The paper is generally difficult to follow. Paragraphs and sentences are not well connected. Furthermore, I advise considering using standard keywords to better present the research. Remove the keywords to ML/DL methods, instead use the standard keywords not more than 5. Please revise the abstract according to the journal guideline. It must be under 200 words. The research question, method, and the results must be briefly communicated. The abstract must be more informative. I suggest having four paragraphs in the introduction for; describing the concept, research gap, contribution, and the organization of the paper. The motivation has the potential to be more elaborated. You may add materials on why doing this research is essential, and what this article would add to the current knowledge, etc. The originality of the paper is not discussed well. The research question must be clearly given in the introduction, in addition to some words on the testable hypothesis. Please elaborate on the importance of this work. Please discuss if the paper suitable for broad international interest and applications or better suited for the local application? Elaborate and discuss this in the introduction.
State of the art needs significant improvement. A detailed description of the cited references is essential. Several recently published papers are not included in the review section. In fact, the acknowledgment of the past related work by others, in the reference list, is not sufficient. Consequently, the contribution of the paper is not clear. Furthermore, consider elaborating on the suitability of the paper and relevance to the journal. Kindly note that references cited must be up to date.
Round 2
Reviewer 1 Report
Recommended
Reviewer 2 Report
OKAuthor Response
Please see the attachment.

Reviewer 3 Report
The authors addressed the comments provided in the first round of reviews.
More specifically, they:
- Grouped paragraphs more cohesively, improving the reading quality of the manuscript.
- Added details on the dataset used in the experiments.
- Include more recent studies involving deep neural networks including 1D-CNN approaches.
- Fixed typos in the paper and improve the readability
- Clarified the unit of measurement reported on the x-axis and the y-axis of the plots
For these reasons I recommend the paper be accepted as is.
Reviewer 4 Report
I think that this new version could be published after some issues are corrected.
Some concepts should be further explained as they are clear only for experts: kernel, padding, stride, ReLU…
The genetic algorithm should be further described.
The outputs provided by the 1D CNN must be described. As they are classes they must be defined: How many were considered? How were they defined? Which is their form (number of neurons in the output and how these outputs codify each class)? Some examples should be provided.
What do the errors in Tables 1-5 mean? Which do they measure: MAPE, MAE, MSE..? How were they measured? In other words: how were compared the predictions provided by the 1D CNN with the desired ones? How were the corresponding errors obtained?
I suggest a last revision of the English language as some phrases are not correct.
Reviewer 5 Report
All comments had been addressed and the manuscript meets the revision requirements.
